# Efficacy of Stromal Vascular Fraction Treatment for Knee Osteoarthritis: A Single-Arm Experimental Trial

**DOI:** 10.3390/biomedicines13122913

**Published:** 2025-11-28

**Authors:** Anna Boada-Pladellorens, Merce Avellanet, Anna Veiga, Esther Pages-Bolibar

**Affiliations:** 1Physical Medicine and Rehabilitation Department, Hospital Nostra Senyora de Meritxell, AD700 Escaldes-Engordany, Andorra; 2Celular Clinic, AD700 Escaldes-Engordany, Andorra; 3Research Group on Health Sciences and Health Services, University of Andorra, AD600 Sant Julià de Lòria, Andorra; 4Barcelona Stem Cell Bank, Regenerative Medicine Programme, Institut d’Investigació Biomèdica de Bellvitge (IDIBELL), Hospital Duran i Reynals, 08908 Barcelona, Spain

**Keywords:** stromal vascular fraction, mesenchymal stem cells, knee osteoarthritis, regenerative medicine, clinical trial

## Abstract

**Background/Objectives:** Knee osteoarthritis (KOA) is a common pathology characterized by impaired joint cartilage. Mesenchymal stromal cell (MSC)-based treatments, such as stromal vascular fraction (SVF), are increasingly being used for their potential cartilage-generating capabilities; however, there is still insufficient evidence to confirm their effectiveness. The aim of the study was to assess the efficacy of SVF treatment in KOA in terms of pain relief. **Methods:** An experimental clinical trial was performed. We included adults with symptomatic KOA who attended Celular Clinic (Andorra). A laboratory-manufactured and standardized SVF product (Celstem^®^) was applied to selected patients. Clinical, functional, and radiological assessments using the visual analog scale, KOOS (Knee Injury and Osteoarthritis Outcome Score), SF-36 scale, and MOCART classification (Magnetic Resonance Observation of Cartilage Repair Tissue) were performed. Variables were compared before treatment and at one, six, and twelve months after treatment. Adverse effects were reported. **Results:** In total, 184 patients were included in the clinical trial, 78 of whom were finally analyzed. There were statistically significant differences in both resting and activity-related pain and in all KOOS subscales after SVF treatment (*p* < 0.001). The quality of life also showed significant changes (*p* = 0.021). No significant changes were observed in MOCART values. However, a positive association was found between MOCART and cell yield. Few adverse effects were reported. **Conclusions:** Our nonrandomized uncontrolled clinical trial showed that SVF treatment has promise to reduce pain in patients with KOA. Improvements in functionality and quality of life were also observed. Future randomized controlled trials regarding SVF versus placebo therapies will further clarify this potential.

## 1. Introduction

Osteoarthritis is one of the primary causes of disability in adults, and the knee is the most commonly affected joint, leading to knee osteoarthritis (KOA). KOA is now recognized as a whole-joint disease rather than a disorder confined to articular cartilage degeneration. It involves pathological alterations across multiple joint tissues, which interact biomechanically and biochemically, contributing to inflammation, pain, and progressive structural damage within the joint [1].

The pathogenesis of KOA reflects the interactions between mechanical stress, inflammatory and metabolic pathways, and disrupted crosstalk among joint tissues. These processes are influenced by a range of risk factors, both modifiable and non-modifiable. Established risk factors include older age, female sex, obesity, and previous joint injury, each contributing to altered joint loading and metabolic dysregulation [2]. Biomechanical factors, such as varus/valgus malalignment, repetitive occupational kneeling, squatting, or heavy physical labor, further increase risk [3]. In addition, emerging evidence links high bone mineral density, metabolic syndrome, and low socioeconomic status with a higher incidence of KOA [4,5].

Collectively, these findings underscore that KOA represents a heterogeneous, whole-joint disease arising from the convergence of systemic, mechanical, and local tissue-specific factors, emphasizing the need for integrated, tissue-targeted therapeutic strategies.

New treatments for KOA include cell therapy with mesenchymal stem cells (MSCs). In arthritis animal models, considerable regeneration and repair in articular cartilage have been observed after MSC transplantation into articular cavities. Treatment with MSCs has also been demonstrated to reduce cartilage lesions by restraining the onset of inflammation [6]. In clinical studies, the effectiveness and safety of treatment with MSCs from various sources for KOA have been previously assessed [7,8]. In addition, MSC treatment has been shown to significantly improve pain and functional status of patients with KOA [9].

MSCs isolated from adipose tissue are genetically and morphologically stable, and they proliferate better over an extended incubation period than other MSC types [10]. Adipose-derived mesenchymal stem cells (ADSCs) and stromal vascular fraction (SVF) derived from adipose tissue are easy to obtain and have higher isolation yields than MSCs from other sources [11]. SVF is a product of heterogeneous composition, predominantly containing ADSCs but also including macrophages, blood cells, pericytes, fibroblasts, and endothelial cells and their progenitors [12]. This cellular heterogeneity endows it with a high therapeutic potential in immunomodulation, anti-inflammatory action, and angiogenesis [13].

The safety of SVF treatments has been widely assessed. However, efficacy remains a concern, as substantially different manufacturing processes are used to produce the SVF product. Although several studies have reported the superiority of SVF over other therapies [14,15,16,17,18], the evidence remains limited by the lack of methodologically rigorous clinical research and the absence of evaluation of anatomical changes using validated scores [18,19]. Moreover, conducting a clinical trial in compliance with the International Standards of Good Clinical Practice [20] presents significant challenges in the field of SVF therapies. These challenges include limited access to experienced cell therapy biologists and specialized laboratories, strict legal and regulatory requirements, the ethical constraints that preclude the use of a placebo, and the high cost associated with cell therapy procedures. These factors hinder the development of robust, high-quality clinical trial protocols. This study sought to evaluate the efficacy of SVF treatments with the highest methodological rigor.

Therefore, we hypothesized that SVF treatment could relieve pain and enhance functionality and quality of life in patients with KOA. The aim of this study was to assess the efficacy of SVF treatment in adult patients with KOA in terms of pain relief. Secondary outcomes included evaluating treatment safety and the effect of SVF on functionality and quality of life, as well as radiological changes after treatment, and the possible correlation of sociodemographic and/or clinical features with SVF treatment.

## 2. Materials and Methods

### 2.1. Study Design and Setting

This study was a prospective, single-arm experimental clinical trial with a one-year follow-up. We conducted the trial in Celular Clinic, a private clinic from Escaldes-Engordany (Andorra) focused on regenerative medicine treatments, between May 2018 and May 2023 (a one-year extension was required due to the COVID-19 pandemic to reach the expected sample size). The study design was selected in alignment with real-world clinical practice, considering that patients are evaluated within a private healthcare setting and are responsible for the cost of their treatment. The protocol of this clinical trial was published in ClinicalTrials.gov (clinical trial number NCT04749758, retrospectively registered, 10 February 2021). The study was approved by the Clinical Research Ethics Committee of Hospital Nostra Senyora de Meritxell, Andorra. After receiving extensive information and once the patient accepted the treatment indication, the patient signed the informed consent and personal data transfer consent (see Appendix A). We followed the Minimum Information for Studies Evaluating Biologics in Orthopedics: Platelet-rich Plasma and Mesenchymal Stem Cells Methodology Guidelines [21]. We conducted the study in accordance with CONSORT guidelines (see Appendix A).

### 2.2. Participants

We included all adult (>18 years old) male and female patients with unilateral or bilateral symptomatic KOA, determined by the American College of Rheumatology diagnostic criteria [22], and showing any degree of KOA in the Kellgren and Lawrence classification [23], who attended the Celular Clinic. Included patients were required to have had KOA symptoms for more than six months, which had to persist despite conventional treatment for KOA. We excluded patients with unclear, diagnosis, or history of active or recent (less than six months) joint infection, pregnant or breastfeeding women, those with neurological deficit in the affected limb, varus or valgus higher than 20 degrees in the affected limb, positive serology for hepatitis B, hepatitis C, and/or HIV, magnetic resonance imaging (MRI) without pathological alterations, and those who had received knee infiltration with glucocorticoids, hyaluronic acid, platelet-rich plasma, and/or other regenerative medicine therapies in the previous three months. We also excluded patients with medical conditions (including the diagnosis of oncological pathology or infectious and severe heart, kidney, or liver disease, which contraindicate sedation or liposuction procedure), history or diagnosis of dementia, and those not having the required skills and/or electronic devices necessary to complete the online questionnaires.

Patients could be withdrawn from the study if they did not complete the questionnaires within the pre-established time frame and in the required format, or if their tracking was lost. The withdrawal of the patients did not represent any prejudice against the patients in the control of their disease or their management by the medical team.

The Principal Investigator (PI) or collaborators collected the patient’s medical history and physical examination during the first medical appointment. Specifically, all surgical and/or intra-articular procedures (with steroids, hyaluronic acid, and/or platelet-rich plasma) were recorded. We performed a physical examination at every follow-up visit.

### 2.3. Intervention

The object under study was a standardized treatment in regenerative medicine. The SVF product developed by Cellab Laboratory (Celstem^®^, Cellab Laboratory, Sant Julià de Lòria, Andorra) is a biological product approved by the Andorra Government authorities for clinical application and strictly manufactured to ensure its reproducibility, quality, and safety. The included patients were treated with an intra-articular injection of Celstem^®^.

Celstem^®^ was obtained through liposuction and was prepared in a laboratory cleanroom, following the American Association of Blood Banks’ conditions of hygiene and biological safety. First, an exhaustive washing of the fat extracted was performed. Then, after adding an enzymatic digestion solution, the mixture was incubated and stirred. Human albumin was used to inactivate collagenase. The final manufacturing step involved thoroughly rewashing the mixture. The product was administered by intra-articular injection under sterile conditions and guided by ultrasonography. The entire procedure was consistently performed by the same professionals (i.e., a plastic surgeon, a biotechnologist, and two physical medicine and rehabilitation physicians) to minimize interpersonal biases (See Appendix A for details on Celstem^®^ manufacturing and administration). We collected intrinsic biological data (e.g., the amount of adipose tissue sample extracted, effective dose, cellularity, immunophenotype, and sterility) for every SVF product to characterize it. For sterility control, cultures were performed on a small sample of both the initial and final products. We carried out a Gram stain in cases of non-sterility to identify the causative microorganism.

### 2.4. Outcomes

The PI and collaborators of this study undertook the recruitment and follow-up visits. The follow-up period lasted one year, with clinical assessments at one, six, and twelve months after the treatment. We assessed primary and secondary outcomes before treatment and at all follow-up visits after treatment, except for MRI, which was performed only before and one year after receiving SVF.

The primary outcome was pain, which was assessed using a visual analog scale (VAS). We evaluated pain both at rest and during activity to better assess the burden of the disease while in motion. The scale ranged from 0 to 10, where 0 meant no pain and 10 meant the worst pain imaginable. We determined minimum clinically important difference (MCID) values [24] according to the absolute value of pain at the beginning of the study: for a VAS ≤ 3.4, the MCID was 1.3; for a VAS between 3.5 and 6.6, the MCID was 1.7 ± 1; and for a VAS ≥ 6.7, the MCID was 2.8 ± 2.1 [25]. Of note, the KOA clinical-radiological dissociation (widely described by many authors [26,27]) prevents the exclusion of patients with low VAS scores to avoid missing opportunities in radiologically advanced patients.

Secondary outcomes included Patient-Related Outcome Measures (PROMs) to determine the effect of SVF treatment on functionality and quality of life. We measured functionality with the validated Spanish version of the Knee Injury and Osteoarthritis Outcome Score (KOOS) scale [28]. We evaluated the quality of life with the validated Spanish version of the SF-36 questionnaire [29], which consisted of a set of questions answered in approximately 10 min to quantify and compare the overall health status of patients. The questionnaire is divided into eight scales, measuring three aspects: functional status, well-being, and a global subjective evaluation of one’s health. Eight scores ranging from 0 to 100 were obtained (the higher the score, the better the health status) [30]. We analyzed radiologic joint morphological changes using MRI in all participants, both before and after SVF treatment, and at one year post-treatment. MRI analysis was performed by one of two radiologists, each with more than 10 years of clinical experience in musculoskeletal radiology. Examinations were performed on a 1.5 T MRI system (Achieva SmartPath to dStream, Philips N.V., Koninklijke, The Netherlands) with a dedicated 16-channel knee antenna. We recorded each patient’s product biological data: whether it was a fresh or frozen sample, the amount of adipose tissue sample extracted (in mL), the amount of concentrated adipose tissue (in mL), the total cellular dose, the effective cell yield dose, the cellular viability (in %), the percentage of SVF immunophenotype, and the sterility. We analyzed sociodemographic and clinical variables that could predict or influence the treatment effects and their correlation to the biological data of the SVF. We collected adverse events or side effects related to the treatment.

### 2.5. Statistical Analysis and Sample Size

The sample size was determined with the Sample Size Calculator GRANMO (Version 7.12, April 2012). A random sample of 77 individuals affected by KOA was expected to be sufficient to estimate, with 95% confidence and a margin of error of ±5 percentage points, an effect of the SVF treatment. The population was expected to be around 5% of the country’s population (similar to the population of the regions of Andorra and Catalonia). An anticipated replacement rate of 5% was also considered [31].

For the statistical analysis, we used descriptive statistics to summarize data with mean and standard deviation (SD) or median and range or interquartile range (IQR). We used the *t*-test and the Mann–Whitney U test for bivariate analyses involving numerical variables, depending on their distribution. We assessed categorical variables with the Chi-square test. We used the Spearman correlation coefficient to analyze the associations between clinical variables, and the paired signed-rank test was applied to analyze intragroup differences between the follow-up time points and the start of the study. We calculated two-sided *p*-values, adjusted by multiplicity using the Holm-Bonferroni method when appropriate, and set the statistical significance level at *p* ≤ 0.05. We carried out all analyses using the RStudio program, Version 4.2, for Microsoft Windows.

## 3. Results

### 3.1. Baseline Demographic and Clinical Characteristics

A total of 184 patients affected by KOA were evaluated at the Celular Clinic center between May 2018 and May 2022. Of these, 45 were excluded from the study because they did not meet the inclusion criteria. As the study was carried out in a private clinic specializing in regenerative medicine, where such treatments constitute the exclusive clinical practice, all patients attending the center were screened as potential candidates for inclusion. Nevertheless, individuals who did not meet the diagnostic criteria for osteoarthritis, or who presented osteoarthritis in joints other than the target site, were excluded from participation. The final number of patients analyzed was 78 (Figure 1). The recruitment process was completed when the expected sample size was reached.

The sample comprised 46 (59.0%) men, and the mean age was 59.12 (SD: 11.27) years. At the time of inclusion, 60 (76.9%) patients had a BMI corresponding to normal weight or mild overweight, while 18 (23.1%) were classified as obese (BMI ≥ 30). Regarding the severity of KOA at baseline, the median MOCART score was 65.0 (range: 10.0–135.0). There was a predominance of osteoarthritis in the lateral femorotibial compartment compared with the medial compartment, with the patellofemoral compartment being the second most affected of the three (Table 1).

At baseline, pain measured with the VAS was lower at rest (median: 1; range: 0–9) than during activity (median: 6; range: 1–10). Other PROMs at baseline are displayed in Table 2.

### 3.2. Pain

There were statistically significant differences in resting and activity-related pain, expressed through the VAS, between baseline values (at rest, median: 1, range: 0–9; in activity, median: 6, range: 1–10) and all other post-treatment evaluation times (1, 6, and 12 months). Notably, pain decreased in the first month after treatment (at rest, median: 0, range 0–7; in activity, median: 3, range: 0–10; *p* < 0.001), and this improvement was maintained after six (median: 2.5, range: 0–8; *p* < 0.001) and twelve months (median: 3, range: 0–9; *p* < 0.001). There were highly significant differences in activity-related pain, which was the most limiting for patients and had higher baseline values (Figure 2).

### 3.3. Functionality

The KOOS total score significantly improved one month after treatment (at baseline, median: 63.1, range: 19.6–91.5 vs. at one month, median: 73.8, range: 39.9–98.8; *p* < 0.001) and remained at higher levels than baseline after 12 months (median: 80.36, range: 22.63–100; *p* < 0.001). All KOOS subscales (symptom, pain, activities of daily living, sport, and quality of life) showed harmonized evolution throughout the follow-up.

### 3.4. Quality of Life

All average values of the eight subscales on the SF-36 either improved or remained constant after treatment. The change was statistically significant for physical functioning (*p* = 0.021), role limitations due to physical health (*p* = 0.023), role limitations due to emotional problems (*p*  =  0.017), energy/fatigue (*p*  =  0.012), social functioning (*p* < 0.001), and pain (*p*  =  0.023). The assessment of perceived health change over time also increased significantly (*p*  =  0.007). Improvements were already significant at the 6-month follow-up. The emotional well-being and general health subscales did not show significant changes throughout the follow-up period (Table 3)

### 3.5. Radiologic Cartilage Changes

Although some patients showed better overall MOCART scores, changes were not statistically significant. Changes observed in thickness, surface, signal intensity, or subchondral bone 12 months after treatment compared with baseline did not reach statistical significance in any knee joints (patellofemoral, lateral, and medial tibiofemoral).

The potential relationship between clinical data and the change in the MOCART score was analyzed. The only notable association observed was between the cell yield contained in the SVF product and the total MOCART score, indicating a potentially relevant positive association (*p* = 0.032). A higher MSC percentage was associated with a greater positive change in the total MOCART score. Additionally, radiological improvements were noted in the MRI scans of several patients (see examples in Figure 3).

### 3.6. Biological Data

The manufacturing certificates for all SVF treatments (Celstem^®^) were analyzed: 76 (97.4%) SVF samples were fresh (extracted, processed, and infiltrated on the same day), while 2 (2.6%) samples were frozen-thawed (1 for reinfusion one year after the first one and 1 used a sample from previously preserved adipose tissue). Neither age nor weight was a significant factor influencing the biological characteristics of SVF preparation in Celstem^®^.

A median of 175 mL (range: 15–500 mL) of adipose tissue was extracted, and once concentrated by centrifugation, a median of 50 mL (range: 10–162 mL) was available. A total of 35.9% of fat samples were non-sterile at the time of extraction; however, this issue was addressed during product manufacturing, resulting in a final contamination rate of 25.6% (20 samples). The contamination source was mainly saprophytic skin flora microorganisms, such as Staphylococcus epidermidis (94.4% of cases), and there was a single case of *Corynebacterium* sp.

The average cellular yield (cells per gram of fat) was 4.26 × 10^5^ cells, with an average of 11.71% of these being compatible with the SVF immunophenotype (Table 4). The relationship between the SVF product cell yield and all clinical variables was evaluated. The variables were categorized into ≤1 million and >1 million cells. Significant differences were found only in one KOOS subscale; however, this result was not clinically relevant. Therefore, a minimum effective cellular dose for the treatment in our study sample could not be established.

### 3.7. Safety Profile

A total of 73.1% of patients reported no adverse effects from the treatment. No incidents related to the anesthetic procedure were recorded. In the postoperative period following liposuction, 26.9% of patients reported minor discomfort at the extraction site, such as edema and/or bruising, with most patients experiencing mild symptoms that resolved spontaneously within a few days. Some patients underwent physiotherapy sessions consisting exclusively of continuous ultrasonography treatment to accelerate abdominal hematoma absorption (with a maximum of 5 sessions). Patients did not report any deformities or irregularities in the abdominal wall, nor did they express dissatisfaction with the esthetic outcome after liposuction. There were no major or minor adverse effects (including pain and infection) related to the intra-articular injection of the SVF product.

### 3.8. Other Analyses

The results for all study variables (pain, functionality, quality of life, and MOCART score) were analyzed by sex, age, BMI, and KOA severity, and no significant differences were found in the outcomes.

## 4. Discussion

The efficacy of SVF in relieving pain and enhancing the functionality and quality of life in adult patients with KOA was evaluated. Our results showed that following SVF treatment, both resting and activity-related pain significantly decreased in KOA patients. Following the therapy, the patient’s functionality and quality of life also improved. In addition, the low rate of adverse effects, which were mild and mostly resolved spontaneously, demonstrated the safety of SVF treatment.

Despite mild-to-moderate KOA initial pain levels, a statistically significant improvement in pain was observed at rest and during activity, consistent with previously reported data [14,32,33,34,35]. The clinical-radiological dissociation in KOA, widely described in the literature [26,27], suggests that pain is an imprecise marker of the radiological severity of KOA, and it must be associated with other variables to achieve an accurate assessment of the patient’s health status. Based on this concept, the selection of patients for our study was not conditioned by a specific VAS score. The determining factor for inclusion was the MRI diagnosis of KOA, regardless of the severity of accompanying clinical symptoms. This approach facilitated the inclusion of patients with different disease stages.

Evaluating health status in KOA also involves considering functionality and quality of life, among other aspects. This approach is the first step in evaluating well-being, disease progression, and the effectiveness of interventions [36]. Statistically significant functional improvement across all five subscales of the KOOS (exceeding the eight-point MCID described by Marot et al. [37]) 12 months after treatment was achieved, in line with previous findings [35,38,39]. On the contrary, quality of life had not been measured in published studies [40]. Statistically significant changes were observed in six of the eight subscales of SF-36 and in health perception over time. Following the SVF treatment, the quality of life of the study population improved, suggesting that SVF might be responsible.

Structural changes in the knee joint have been sought after over the years. Although SVF has emerged as a potential treatment for cartilage regeneration because of its in vitro effects, evaluating anatomical changes using validated scores has been rarely undertaken [19]. In spite of the insufficient statistical power of our study, a weak yet potentially relevant positive correlation between the percentage of MSCs and improvement in the MOCART classification should be highlighted. Higher percentages of MSCs were associated with a more significant positive change in the MOCART total score. Song et al. [41] reported similar findings after three ADSC infiltrations over 96 weeks. Our results also suggest a similar trend with a single infiltration, indicating a potentially greater regenerative capacity of SVF products than that of purely expanded MSCs. While Muthu et al. [42] observed superior functional outcomes with non-cultivated MSCs for KOA compared with cultivated MSCs, the debate continues with compelling arguments supporting both cultivated and non-cultivated products. More extensive studies with larger patient populations are needed to establish clear associations and provide more precise guidance on treatment indications, types, and timing, thereby enhancing therapeutic efficacy and regenerative potential.

To date, no widely recognized reproducible manufacturing procedure for SVF treatments exists. There are different commercially available kits, but despite their similar appearance, they differ in the permitted digestion time with collagenase, the neutralization process, centrifugation time and intensity, SVF cellular yield, viability, composition, cost, and total processing time. Given this heterogeneity, although some authors advocate using kits over the reference method (manually produced in a laboratory by a technician), [43,44] the best way to obtain SVF is still debated. In our study, using a standardized product with a final manufacturing certificate enabled consistent analysis of clinical and biological results.

The minimum effective dose for cellular therapy remains undetermined. Some authors have suggested that the minimum number of MSCs required to produce the desired therapeutic effect is 2 × 10^6^ cells per kilogram of body weight [45]. However, the dose and frequency of MSCs needed may vary depending on the severity of the disease [46]. Some studies (mainly involving expanded MSCs) have found a relationship between cell dose and clinical outcomes by categorizing cell doses, but a therapeutic threshold is yet to be definitively established [45,46]. In our study, the relationship between the number of cells in the SVF product and all clinical variables was analyzed. Despite categorizing this variable into two groups (≤1 M and >1 M cells), a clear minimum effective cellular dose could not be observed.

In addition, the analysis of SVF clinical and biological data did not allow for identifying the patient profile that could benefit the most from treatment. Previous clinical studies analyzing MSC treatments have shown favorable clinical results in the advanced and early stages of KOA [46]. However, considering our results as a whole, our observations support a potential therapeutic role of SVF therapy in patients with KOA within the “treatment gap” [47]. As London et al. described it, “This period spans from the exhaustion of ineffective conservative treatments to surgical intervention and is considered non-benign. It often represents a long duration—years, and frequently decades—during which patients experience debilitating pain, a reduction in quality of life, and significant financial burden. The reasons for this period likely include the inefficacy of conservative measures in long-term symptom management, the lack of safe and effective minimally invasive treatments, and a significant reluctance or unwillingness of patients to undergo major, irreversible surgeries like arthroplasty or knee osteotomy, particularly in younger osteoarthritis patients” [47]. Iolascon et al. suggested that detecting KOA at an earlier stage could facilitate a better prescription of innovative therapeutic approaches, such as SVF [48]. The long-term effects of this therapy in our cohort of patients with early-stage osteoarthritis would be of great clinical interest, as they would help determine the optimal timing of treatment based on disease stage.

Regarding safety, our study confirmed the low rate of mild adverse effects previously described [19]. Notably, all adverse effects were related to the extraction procedure, and none were associated with the knee joint.

The generalizability of our findings is limited by several factors. First, although MSCs are the focus of current research in some medical fields, many properties and the therapeutic potential of their subtypes remain unknown. While their efficacy in modulating inflammation has been demonstrated in various animal models, the results obtained in human clinical trials have been more modest [49]. Several controversial issues persist regarding their biology (including their specific phenotype, the need for an inflammatory environment to induce immunosuppression, and the cell delivery route, among others). Therefore, a consensus on the definition, composition, mechanism of action, and manufacturing process of MSCs is yet unavailable. Second, our study had a living lab approach [50]. As an exploratory study in a private clinic, neither the placebo nor the randomization process was considered. This may lead to potentially misleading or inaccurate conclusions, due to the placebo and regression to the mean effects of KOA [51]. Third, we included all grades of KOA, which could mask heterogeneous treatment effects. Finally, and for ethical reasons, our study did not include histological analysis of the articular cartilage of enrolled patients. Although some authors have described the immunomodulatory and cartilage-regenerating effects of SVF in vitro [52], the duration that SVF cells maintain their intra-articular properties remains unknown. Our study on humans provided evidence that the treatment improved clinical features; however, we could not assert that these improvements corresponded to new articular cartilage with properties similar to those of the original cartilage.

## 5. Conclusions

In conclusion, our clinical trial showed that SVF treatment may reduce pain in patients with KOA. Following SVF therapy, functionality and quality of life in patients evaluated one, six, and twelve months after treatment significantly improved. Although statistically significant differences were not found, SVF treatment may have had a positive influence on the radiological progression of KOA, with notable improvements observed on MRI one year after treatment. The therapy was safe, and only a few minor reversible adverse effects were reported. Further randomized controlled clinical trials with larger samples comparing exclusively standardized SVF treatment with a control group are necessary to identify which subgroup of patients may have the greatest regenerative response to treatment.

## Figures and Tables

**Figure 1 biomedicines-13-02913-f001:**
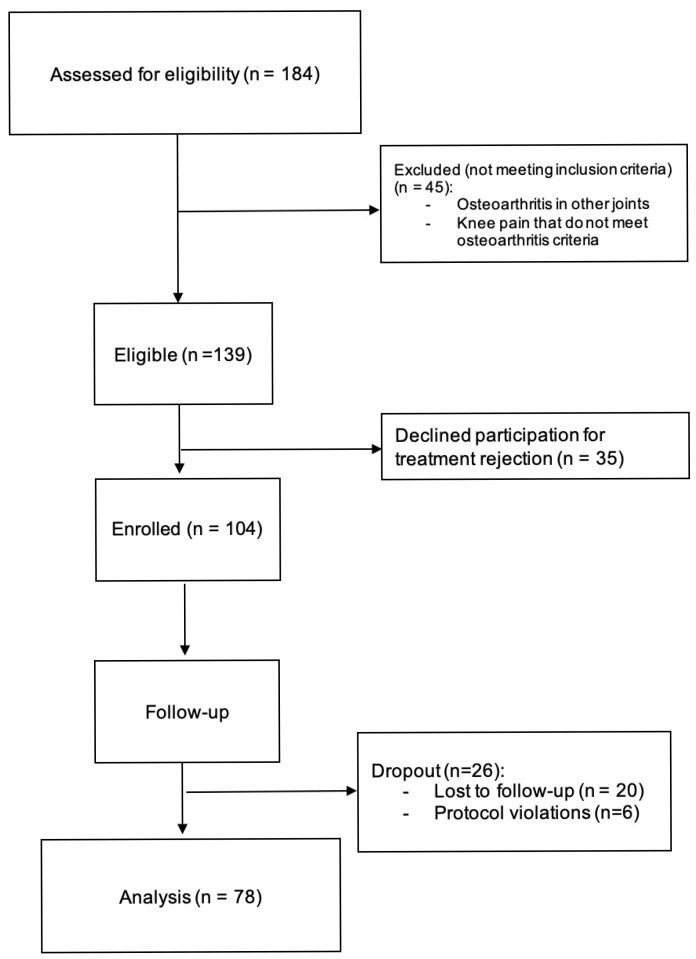
Flow diagram of the study participants’ selection.

**Figure 2 biomedicines-13-02913-f002:**
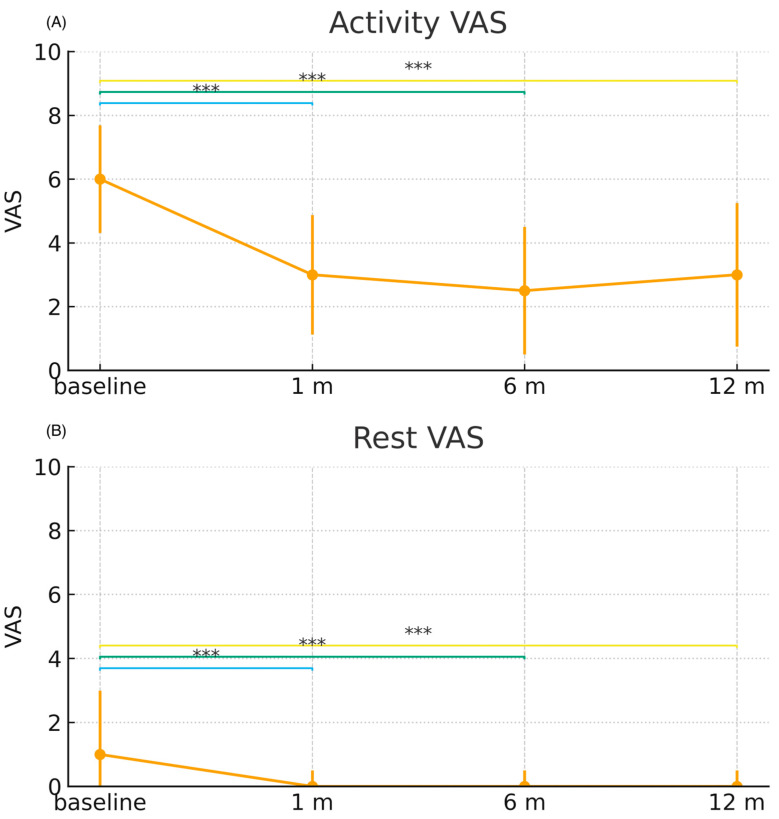
Changes in Visual Analog Scale (VAS) pain scores over time following treatment. Panel (**A**) shows VAS values during activity, and Panel (**B**) shows VAS values during rest. Data are presented as median and interquartile range (IQR) at baseline, 1 month, 6 months, and 12 months after treatment. Statistical comparisons were performed using the paired nonparametric Wilcoxon signed-rank test within the treatment arm. Significance levels are indicated as: *** *p* < 0.001; ns = not significant.

**Figure 3 biomedicines-13-02913-f003:**
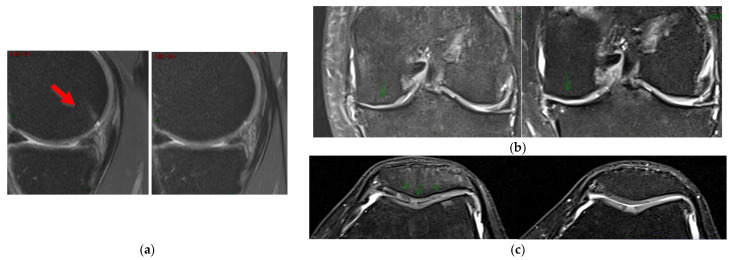
MRI images pre- (**left**) and post-treatment (**right**). (**a**) A decreased signal intensity and surface extension in subchondral bone were found (red arrow). (**b**,**c**) An increased cartilage thickness was found in the medial tibiofemoral joint and in the patellofemoral joint, respectively (green arrows).

**Table 1 biomedicines-13-02913-t001:** Baseline demographic and clinical characteristics of study participants.

	Study Sample (N = 78)
Sex, No. (%)	
Female	32 (41.0)
Male	46 (59.0)
Age (years), mean (SD)	59.12 (11.27)
<60 years, No. (%)	42 (53.8)
≥60 years, No. (%)	36 (46.2)
BMI, mean (SD)	27.05 (3.95)
<30, No. (%)	60 (76.9)
≥30, No. (%)	18 (23.1)
Affected knee, No. (%)	
Left	39 (50.0)
Right	39 (50.0)
Reinfusion, No. (%)	
Yes	8 (10.3)
No	70 (89.7)
Previous treatments, No. (%)	
Platelet-rich plasma	23 (29.5)
Steroids	9 (11.5)
Hyaluronic acid	16 (20.5)
Surgery ^a^	27 (77.1)
MOCART ^b^, median [range]	
Lateral tibiofemoral joint	30.0 [0.0–50.0]
Medial tibiofemoral joint	15.0 [0.0–50.0]
Patellofemoral joint	20.0 [0.0–50.0]
Joint effusion	0.0 [0.0–5.0]
Total	65.0 [10.0–135.0]

^a^ Data available for 35 participants. ^b^ Data available for 75 participants. Abbreviations: BMI, body mass index; MOCART, magnetic resonance observation of cartilage repair tissue; SD, standard deviation.

**Table 2 biomedicines-13-02913-t002:** Outcome data at baseline.

	Study Sample (N = 78)
VAS ^a^, median [range]	
Rest	1 [0–9]
Activity	6 [1–10]
KOOS ^b^, median [range]	
Pain	63.9 [25.0–100]
Symptom	75.0 [28.6–100]
Activities of daily living	75.0 [5.8–98.5]
Sport	30.0 [0–95.0]
Quality of life	31.3 [0–75.0]
Total	63.1 [19.6–91.5]
SF-36 ^c^, median [range]	
Physical functioning	57.5 [0–100]
Role limitations due to physical health	50 [0–100]
Role limitations due to emotional problems	100 [0–100]
Energy/fatigue	62.5 [20–100]
Emotional well-being	78.3 [10–100]
Social functioning	87.5 [12.5–100]
Pain	55 [0–100]
General health	72.5 [15–100]
Health change	50 [0–100]

^a^ Data available for 74 participants. ^b^ Data available for 71 participants. ^c^ Data available for 60 participants. Abbreviations: KOOS, knee injury and osteoarthritis outcome score; SF-36, short-form 36; VAS, visual analog scale.

**Table 3 biomedicines-13-02913-t003:** Changes in quality of life between baseline and each follow-up according to the SF-36 questionnaire, Median [IQR].

	BS	1 m	*p*-Value ^a^	6 m	*p*-Value ^a^	12 m	*p*-Value ^a^
Physical functioning	57.5 [50–71.25]	72.5 [50–85]	0.007	75 [50–87.5]	<0.001	75 [50–87.5]	0.021
Role limitations due to physical health	50 [0–100]	50 [0–100]	0.350	100 [50–100]	<0.001	100 [18.75–100]	0.023
Role limitations due to emotional problems	100 [66.67–100]	100 [100–100]	<0.001	100 [100–100]	<0.001	100 [100–100]	0.017
Energy/Fatigue	62.5 [45–80]	67.5 [55–80]	0.509	70 [60–82.5]	0.026	77.5 [60–85]	0.012
Emotional well-being	78.33 [52–90]	80 [65–89.5]	1	80 [70–90]	1	80 [70–90]	1
Social Functioning	87.5 [72–100]	100 [75–100]	0.094	100 [87.5–100]	<0.001	100 [75–100]	<0.001
Pain	55 [35–70]	67.5 [45–78]	0.117	77.5 [45–90]	<0.001	77.5 [45–90]	0.023
General Health	72.5 [56.2–83.7]	75 [55–83.3]	1	75 [56.2–83.3]	1	75 [56.2–83.3]	1
Health change	50 [25–50]	50 [50–75]	0.021	75 [50–75]	<0.001	50 [50–75]	0.007

^a^ Compared with baseline values using the paired nonparametric signed-rank test. Abbreviations: BS, baseline; m, month; IQR, interquartile range.

**Table 4 biomedicines-13-02913-t004:** Stromal vascular fraction treatment characteristics (N = 78).

Initial Sample, No. (%)	
Fresh	76 (97.4)
Frozen	2 (2.6)
Initial adipose tissue (mL), median [range]	175 [15–500]
Concentrated adipose tissue (mL), median [range]	50 [10–162]
Total dose (log-10 scale), mean (SD)	5.97 (0.53)
Cell yield (log-10 scale), mean (SD)	4.26 (0.53)
Viability (%), mean (SD)	88.67 (9.59)
Stromal vascular fraction (%), median [range]	11.71 [0.36–33.20]
Sterility, No. (%)	
At extraction	50 (64.1)
After manufacturing	20 (25.6)
Adipose tissue	27 (43.5)

Abbreviations: SD, standard deviation.

## Data Availability

The datasets generated and/or analyzed during the current study are not publicly available but are available from the corresponding author upon reasonable request and with permission of Cellab Laboratory.

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
