# Peer review of "Efficacy of Stromal Vascular Fraction Treatment for Knee Osteoarthritis: A Single-Arm Experimental Trial"

_biomedicines, 2025, doi:10.3390/biomedicines13122913_

Round 1
Reviewer 1 Report
Comments and Suggestions for Authors
Title: Appropriate.
Abstract:
Conclusions:
1.RE: page 1, Lines 31-35: “Conclusions: Our clinical trial showed that SVF treatment effectively reduced pain in patients with KOA. Improvements in functionality and quality of life were also observed. SVF treatment tended to positively influence the radiological progression of KOA, although no statistically significant differences were found.”
This is definitely an overstatement.
Change to something like this: “Conclusions: Our nonrandomized uncontrolled clinical trial showed that SVFtreatment has promise to reduce pain in patients with KOA. Improvements in functionality and quality of life were also observed. Future randomized controlled trials regarding SVF vs placebo therapies will further clarify this potential.”
Introduction: Appropriate
Methods: Appropriate.
Results: Appropriate.
- RE: Discussion: Page 3, lines 325-327: “Our results showed that SVF treatment reduced both resting and activity-related pain in KOA patients. The therapy also significantly improved the patient’s functionality and markedly enhanced their quality of life.“
This is an overstatement. Passage of time, exercise, or needling of the joint may have caused these changes. Please change to something like this:
“Our results showed that following SVF treatment both resting and activity-related pain significantly decreased in KOA patients. Following the therapy the patient’s functionality and quality of life also improved.”
- RE: Page 4 lines 346-347: “For the first time, our results showed that the quality of life of the study population was positively influenced by SVF treatment.”
Again, this is an overstatement with same criticism as above. Please change to something like this:
“Following the SVF treatment, the quality of life of the study population improved suggesting that SVF might be responsible.”
- RE: Page 5, lines 413-414: “Because of ethical concerns, neither the placebo nor the randomization process could be used. “
This is complete nonsense. The placebo could have been nothing at all, needling, saline, steroid, NSAIDS, or hyaluronate, all of which been shown to improve KOA. You should probably state that “this was an exploratory study that demonstrated SVF might improve KOA with a low adverse event profile”
- RE: Page 5, lines 414-415: “Third, the sample size of our study was sufficiently large to draw conclusions about the efficacy of SVF treatment,”
This is again untrue. This was not a controlled study thus no conclusions can be drawn as to the efficacy of SVF therapy. Please remove.
In the limitations section, please discuss and review that KOA studies have a powerful placebo effect and regression to the mean effects estimated to be approximately 70%, thus, the studies must be controlled and randomized to come to any conclusions. Time, needling, saline, exercise, weight loss, medications, and psychologic therapy all improve KOA – and any new therapy must be controlled for these. Here are some references on the powerful pacebo effect in KOA that could explain all of the author’s findings:
Placebo Effect and Regression to the Mean in Osteoarthritis
Zhang W. The powerful placebo effect in osteoarthritis. Clin Exp Rheumatol. 2019 Sep-Oct;37 Suppl 120(5):118-123. Epub 2019 Oct 15. PMID: 31621561.
https://www.clinexprheumatol.org/abstract.asp?a=14758
Zhang W, Robertson J, Jones AC, Dieppe PA, Doherty M. The placebo effect and its determinants in osteoarthritis: meta-analysis of randomised controlled trials. Ann Rheum Dis. 2008 Dec;67(12):1716-23. doi: 10.1136/ard.2008.092015. Epub 2008 Jun 9. PMID: 18541604.
https://ard.bmj.com/content/67/12/1716.long
Neogi T, Colloca L. Placebo effects in osteoarthritis: implications for treatment and drug development. Nat Rev Rheumatol. 2023 Oct;19(10):613-626. doi: 10.1038/s41584-023-01021-4. Epub 2023 Sep 11. PMID: 37697077; PMCID: PMC10615856.
Lunde SJ, Vase L, Hall KT, et al. Predicting placebo analgesia responses in clinical trials: where to look next? A meta-analysis of individual patient data. Pain. 2025;166(10):e314-e321. doi:10.1097/j.pain.0000000000003615
Borst JM, Ruoss S, Palmer I, Smith T, Kalunian K, Ward SR. Placebo Effect Sizes in Clinical Trials of Knee Osteoarthritis Using Intra-Articular Injections of Biologic Agents. Arthritis Care Res (Hoboken). 2025;77(8):998-1006. doi:10.1002/acr.25526
Englund M, Turkiewicz A. Regression to the mean for physical function and quality of life in clinical trials for symptomatic knee osteoarthritis. Osteoarthritis Cartilage. 2025;33(3):391-395. doi:10.1016/j.joca.2024.11.006
https://www.oarsijournal.com/article/S1063-4584(24)01473-0/fulltext
Yu SP, van Middelkoop M, Deveza LA, et al. Predictors of Placebo Response to Local (Intra-Articular) Therapy In Osteoarthritis: An Individual Participant Data Meta-Analysis. Arthritis Care Res (Hoboken). 2024;76(2):208-224. doi:10.1002/acr.25212
Fazeli MS, McIntyre L, Huang Y, Chevalier X. Intra-articular placebo effect in the treatment of knee osteoarthritis: a survey of the current clinical evidence. Ther Adv Musculoskelet Dis. 2022;14:1759720X211066689. Published 2022 Jan 31. doi:10.1177/1759720X211066689
https://pmc.ncbi.nlm.nih.gov/articles/PMC8808023/pdf/10.1177_1759720X211066689.pdf
- RE: Conclusion, page 5,lines 424-428:
“In conclusion, our clinical trial showed that SVF treatment effectively reduced pain in patients with KOA. This therapy significantly improved functionality and quality of life in patients evaluated one, six, and twelve months after treatment. Although statistically significant differences were not found, SVF treatment had a positive influence on theradiological progression of KOA, with notable improvements observed on MRI one year
after treatment.”
Again an overstatement not supported by the data. Please change to something like this:
“In conclusion, our clinical trial showed that SVF treatment may reduce pain in patients with KOA. Following SVF therapy functionality and quality of life in patients evaluated one, six, and twelve months after treatment significantly improved. Although statistically significant differences were not found, SVF treatment may have had a positive influence on the radiological progression of KOA, with notable improvements observed on MRI one year after treatment.”
Reviewer 2 Report
Comments and Suggestions for Authors
Mesenchymal stromal cell (MSC)-based treatments, such as stromal vascular fraction (SVF), are increasingly being used for their potential cartilage generating capabilities; however, there is still insufficient evidence to confirm their effectiveness. The aim of the study was to assess the efficacy of SVF treatment in KOA in terms of pain relief. The clinical trial showed that SVF treatment effectively reduced pain in patients with KOA. Improvements in functionality and quality of life were also observed. SVF treatment tended to positively influence the radiological progression of KOA, although no statistically significant differences were found.
- In Line 98-99, “This study was a prospective, single-arm experimental clinical trial with a one-year follow-up” The study design is a single-arm trial lacking a control group (e.g., standard care), which limits causal inference regarding SVF efficacy. The justification for choosing a single-arm design is insufficiently explained.
- In Line 214-216, “A total of 184 patients affected by KOA were evaluated at the Celular Clinic centre between May 2018 and May 2022. Of these, 45 were excluded from the study because they did not meet the inclusion criteria. The final number of patients analysed was 78 (Figure 1).” High patient attrition (184 screened, 78 analyzed) risks selection bias. Reasons for exclusion/attrition are not thoroughly discussed, and potential bias is unaddressed.
- In Line 112-113, “We included all adult (> 18 years old) male and female patients with unilateral or bilateral symptomatic KOA, determined by the American College of Rheumatology diagnostic criteria [23]” Broad inclusion criteria (e.g., varying OA severity) may mask heterogeneous treatment effects. Subgroup analyses (e.g., by Kellgren-Lawrence grade) are lacking.
- In Line 199-200, “The population was expected to be around 5% of the country’s population (similar to the population of the regions of Andorra and Catalonia).” The 5% population prevalence assumption for sample size calculation lacks citation, weakening methodological rigor.
- In Line 202-204, “For the statistical analysis, we used descriptive statistics to summarize data with mean and standard deviation (SD) for normally distributed quantitative variables and median and range or interquartile range (IQR) when normality could not be assumed” Normality testing methods (e.g., Shapiro-Wilk test) are not reported, undermining the choice of parametric vs. non-parametric tests.
- In Line 233-236, “There were statistically significant differences in resting and activity-related pain, expressed through the VAS scale, between baseline values (at rest, median: 1, range: 0-9; in activity, median: 6, range: 1-10) and all other post-treatment evaluation times (1, 6, and 12months).” Repeated measures across time points (baseline, 1, 6, 12 months) increase Type I error risk due to unadjusted multiple comparisons.
- In Line 321-322, “The results for all study variables (pain, functionality, quality of life, and MOCART score) were analyzed by sex, and no significant differences were found in the outcomes” Subgroup analysis is limited to sex, ignoring potential effect modifiers like age, BMI, or OA severity.
Reviewer 3 Report
Comments and Suggestions for Authors
The article «Efficacy of stromal vascular fraction (SVF) treatment for knee osteoarthritis: a single-arm experimental trial” is well written and provides interesting results.
I ask authors to perform some minor changes:
In the abstract and conclusion authors state that SVF treatment tended to positively influence the radiological progression of KOA. I recommend that authors write, that they noted no significant association between SVF treatment and progression of KOA. I think that only significant results should be mentioned in the Conclusion section of the Abstract and the text. See also your results presented at line 264-268.
Did you observe any postoperative knee joint infections or knee joint effusions?
Please be humble and replace the words “cases” and “subjects” by “patients” in the entire manuscript.
Please disclose the treatment costs of one SVF treatment in US-Dollar.
Please comment if you used (not used) large language modules for creation of this manuscript.
Line 15: replace “decreased” by “impaired”.
Line 26: replace “184 patients” by “In total, 184 patients”.
Line 48: replace “interplay” by “interactions”.
Line 117: replace “suspicion” by “unclear”.
Line 119: replace “var” by “varus”.
Line 126: delete “or higher functions unpreserved, which interfere with communication”.
Line 136: replace “detailed” by “recorded”.
Line 426-429: delete “Although statistically significant … one year after treatment.”.
Round 2
Reviewer 1 Report
Comments and Suggestions for Authors
The authors did a credible job of the revising the manuscript.
